# BN-FLEMO$_\Delta$: A Bayesian Network-based Flood Loss Estimation Model for Adaptation Planning in Ho Chi Minh City, Vietnam

Kasra Rafiezadeh Shahi[1,2], Nivedita Sairam[1], Lukas Schoppa[1], Le Thanh Sang[3], Do Ly Hoai Tan[3], and Heidi Kreibich[1]

[1]Section Hydrology, GFZ German Research Centre for Geosciences, Potsdam, Germany
[2]Planetary Boundaries Science Lab, Earth System Analysis, Potsdam Institute for Climate Impact Research (PIK), Member of the Leibniz Association, Potsdam, Germany
[3]Southern Institute of Social Sciences, Ho Chi Minh city, Vietnam

**Correspondence:** Kasra Rafiezadeh Shahi (Kasra.Rafiezadeh.Shahi@pik-potsdam.de)

**Abstract.** The risk of flooding is on the rise in Delta cities, such as Ho Chi Minh City (HCMC) in Vietnam, with projections indicating further increases due to climate change and urbanization. Flood risk analyses, for which loss modeling is a key component, play a crucial role in decisions on flood risk management and urban development. Probabilistic multi-variable loss models are increasingly being used to improve loss estimation, as they describe loss processes better and inherently
provide a quantification of uncertainties. However, such models are often based on input variables that are determined by expert judgment. Thus, we propose the first probabilistic multi-variable flood loss model designed for residential buildings in delta cities such as HCMC (BN-FLEMO$_\Delta$). BN-FLEMO$_\Delta$ is built upon new building-level empirical survey data. The model is developed with an automatic machine learning-based (ML) feature selection framework and a systematic learning process to determine the optimal structure of the Bayesian Network. Based on a methods comparison, we demonstrate the
following key advantages of BN-FLEMO$_\Delta$: 1. enhanced, empirically-based description of flood loss processes leading to improved accuracy in loss estimation; 2. provision of a probability distribution of losses and inherent quantification of modeling uncertainty; 3. network structure allows model application even when data for one or more input variables are missing, which is particularly valuable in data-scarce environments. We therefore expect that BN-FLEMO$_\Delta$ will significantly improve risk analyses in HCMC and similar delta cities and support decision-makers in developing sustainable flood risk management
strategies for these dynamic flood-prone regions.

## 1 Introduction

The risk of flooding in delta cities such as Ho Chi Minh City (HCMC) in Vietnam is severe, due to complex, compound flood situations and rapidly increasing exposure due to population growth, urban sprawl and densification (Garschagen and Romero-Lankao, 2015; Bangalore et al., 2019; on Climate Change , IPCC). HCMC located at the periphery of the Mekong
Delta, experiences pluvial, riverine, and coastal floods (Luu et al., 2019; Vachaud et al., 2019; Nguyen et al., 2021; UNDRR, 2022). The city has 2,953 canals, predominantly sourced from the Dong Nai, Saigon, and Vam Co Rivers. Elevated water levels in rivers and canals, particularly those associated with the Mekong River basin, contribute significantly to widespread

inundation (Nguyen et al., 2023; Cao et al., 2021). Furthermore, heavy and recurrent rainfall exacerbates flooding in HCMC, particularly when it surpasses the drainage system's capacity (Luu et al., 2019). The intricate interplay between topography and land subsidence also significantly influences flood dynamics. Specifically, elevation gradients and surface roughness dictate the flow pathways of floodwaters, while land subsidence, driven by factors such as groundwater extraction and urban development, worsens flooding by raising relative sea levels and altering drainage patterns (Bank, 2010). As a result, it is expected that flood risk will continue to increase in HCMC due to climate change, e.g., increased precipitation and sea level rise, as well as socio-economic changes (Bank, 2010; Hanson et al., 2011; Hallegatte et al., 2013; Lasage et al., 2014; Cao et al., 2021). To counteract this trend, sustainable adaptation strategies need to be planned on the basis of comprehensive risk analyses including reliable flood loss modeling (Apel et al., 2009; Poljansek et al., 2017).

In flood risk analyses, flood loss models play a pivotal role in assessing the impact of hazards on exposed assets such as buildings. Traditionally, deterministic stage-damage functions (SDF-Det), differentiated by building or land use, have been used, with water depth being the sole input variable (Smith, 1994; Merz et al., 2010). However, despite their limitations, SDF-Det is still the most common method for estimating flood-related financial losses and can still be considered state-of-the-art (Scawthorn et al., 2006; Thieken et al., 2008; Schoppa et al., 2020). Nonetheless, with many studies recognizing that damage processes are driven by various factors, such as inundation duration, contamination of floodwater, effectiveness of flood warnings, precautionary measures, etc., multi-variable flood loss models were developed (Wind et al., 1999; Penning-Rowsell and Green, 2000; Thieken et al., 2005; De Moel et al., 2015; Gerl et al., 2016). For instance, Thieken et al. (2008) proposed utilizing various loss-influencing variables, including building type and precautionary measures, in addition to inundation depth, as predictors for a rule-based flood loss estimation model designed for the private sector. Their research demonstrated that such multi-variable models describe damage processes better and thus outperform SDF-Det, but the uncertainties associated with loss estimation remained high (Thieken et al., 2008; Elmer et al., 2010). In addition, the selection of input variables for the loss models heavily relies on literature review and expert judgment. Therefore, advanced, data-based, and automated frameworks for feature selection are crucial to enhance our understanding of complex flood loss processes and to improve flood loss models.

Flood loss modeling is associated with high uncertainty, which can be separated into aleatoric uncertainty that is not reducible, and epistemic uncertainty which can be reduced by more knowledge. Aleatory uncertainty refers to stochastic processes that are inherently variable in time, space, or populations of objects, such as a tree trunk that may severely damage one building and may spare the adjacent building, or localized high-flow velocity that may scour the foundation of one building leading to collapse, whereas a neighboring building may only be inundated (Merz and Thieken, 2009). Epistemic uncertainty results from incomplete knowledge and is related to our inability to understand, measure, and describe the damage processes, and is thus linked to disregarding factors influencing damage or the misjudgment of their manifestations and effects (Merz and Thieken, 2009). It is, therefore, crucial to quantify uncertainties in flood loss estimates and hereby support informed and robust decision making (De Brito and Evers, 2016; Pappenberger and Beven, 2006). To quantify this uncertainty, probabilistic loss models were developed (e.g., Bayesian networks), which inherently provide quantitative information on uncertainty associated both with the random heterogeneity of input data and model structure (Schröter et al., 2014; Vogel et al., 2018; Paprotny et al., 2021). Bayesian networks can capture the joint probability distribution of all input variables and model the probabilistic

dependency among the variables (Jensen and Nielsen, 2007). As such they can better model damage processes and are applicable even when data for one or more predictors are missing. Further, Bayesian networks offer the option to incorporate prior information (e.g., from previous studies or expert knowledge) and communicate the model functionality transparently through the graph structure (Vogel et al., 2014). Nevertheless, probabilistic loss models are also specific to the region and the type of flood for which they were developed (Schröter et al., 2014; Wagenaar et al., 2018; Mohor et al., 2021). The transfer of damage models in time and space is critical and leads to significantly increased uncertainty (Wagenaar et al., 2018; Mohor et al., 2021). This is due to differences in hazard characteristics, e.g. between slowly rising river floods in the lowlands and flash floods in the mountains, as well as differences in vulnerability between countries, e.g., due to differences in building stock. For example, Chinh et al. (2017) showed that under the specific flooding conditions in the Mekong Delta, with relatively well-adapted households, long flood duration, and shallow water depth, water depth does not determine flood damage as much as in other regions. Despite the high flood risk in the Mekong Delta, not many studies focused on flood loss modeling in this region (Luu et al., 2019). Scussolini et al. (2017); Couasnon et al. (2022) used deterministic depth-damage curves to calculate flood risk. Wu et al. (2021) assessed flood risk in HCMC using a natural-human framework, which integrated social and environmental factors to develop flood hazard and vulnerability maps.

The objective of our study is to fill this gap and tackle some of the described challenges in developing the first probabilistic flood loss model tailored for HCMC and similar delta cities. Our framework shall not only enable the quantification of model uncertainty through Bayesian inference, essential for decision-making but also incorporate an automatic feature selection workflow. The latter aids in automatically identifying the key factors determining flood loss in HCMC.

The remainder of the paper is organized as follows. Section 2.1 outlines the survey data utilized for model development. The method is detailed in Section 2.2. Section 3 presents the obtained results along with a discussion. Finally, Section 4 summarizes the conclusions drawn from the study.

## 2 Data and Method

### 2.1 Household Survey Data

The empirical data used in the development of the flood loss model was collected from a questionnaire-based structured household survey with 1000 households (Vishwanath Harish et al., 2022). The households were interviewed face-to-face. The pre-requisite for the households to participate is that they must have experienced flood damages in the last 10 years before 2020, i.e., from 2010. In addition to information pertaining to flood hazard and impacts, the questions additionally covered a variety of topics including household composition, building characteristics, implementation of private precautionary measures, socio-economic aspects, previous flood experience, and their perception of changing flood risk and potential adaptation options. Based on discussions with flood risk experts from HCMC, the survey areas were chosen such that they spread over different socio-economic profiles and flood types. The spatial distribution of the survey can be found in the appendix (see Figure A1). However, the household selection within the areas was made at random. Households were asked to report on two flood events (i.e., the most recent event and the most serious event in terms of impact within the last 10 years). Out of 1000 households, 530

provided information on both types of events, while the remaining households reported only one event (in these cases, the recent event was also the most serious). This resulted in 1530 records of flood loss data. Among these records, 467 contained missing values in one or more of the flood loss predictors (e.g., water depth, inundation duration) or target variable (rloss). To ensure the integrity of the analysis, we adopted a complete-case approach by excluding all records with missing values. Subsequently, 16 loss-influencing variables were selected based on an extensive literature review and consultations with domain experts in Ho Chi Minh City (Table 1). As shown in Table 1, these 16 variables cover a wide range of loss-influencing variables, including hydrological factors, socio-economic indicators, and building characteristics. As for the target variable, the relative loss to the building is computed as the ratio of absolute monetary building damage to the building's reconstruction cost. Further details on these 16 variables can be found in the Appendix (see Table A1).

## 2.2 Development of the Bayesian Network-Based Flood Loss Estimation Model

Our proposed framework consists of two phases: 1. an automatic feature selection process to identify the most important variables that determine flood loss; and 2. development of the Bayesian network-based flood loss estimation model (BN-FLEMO$_\Delta$) (see Figure 1). In the following sections, we elaborate on each of these phases.

### 2.2.1 Automatic Feature Selection Using Machine Learning

The first step of flood loss model development is the identification of the most important factors influencing flood loss. Traditionally, this was done relying on expert judgment or existing literature. In recent years, there has been a shift towards data-driven approaches to identify significant predictors from empirical datasets (Vishwanath Harish et al., 2022; Schoppa et al., 2020; Vogel et al., 2018). Such data-driven approaches provide users with insights into the importance of loss-influencing variables. We propose an automatic feature selection framework that employs the SelectKBest technique to assign scores to each predictor based on their F-statistics (Ayyanar et al., 2022). F-statistics assesses the overall significance of the relationship between independent variables and the dependent target variable and is calculated as the ratio of two variances, one capturing the explained variability by the model, and the other representing unexplained variability (Fisher, 1970). A high F-value signifies a strong relationship between predictors and the outcome, indicating improved model performance. Conversely, a low F-value suggests the model may not substantially enhance predictions (Desyani et al., 2020). SelectKBest then selects the top $k$ predictors with the highest scores for further analysis. Although SelectKBest contributes to improving the predictive performance of models, there are some disadvantages (Ayyanar et al., 2022).

Firstly, the method investigates each predictor variable individually. Secondly, the user must define the parameter $k$ prior to feature selection. As depicted in Figure 1, to address the former challenge, we analyze the predictive performance of selected $k$ predictors using standard multi-variable linear regression (MLR) in terms of mean squared error (MSE) (Hocking, 1976). To address the latter challenge, we create an automatic framework using the grid search technique. This technique serves as a hyperparameter optimizer and searches within a specified range to determine the optimal value of a hyperparameter (Liashchynskyi and Liashchynskyi, 2019). In our study, we explore a range of values for $k$ from 1 to $\mathcal{D}$, where $\mathcal{D}$ represents the original number of features (i.e., 16 features as described in Table 1). By leveraging the proposed automated feature selec-

**Table 1.** The 16 potential flood loss-influencing variables along with the target variable (rloss).

| Categories | Explanatory variables | Type, range, unit | Mean, Median |
|---|---|---|---|
| Hydrologic aspects | Water depth (wd) | Continuous, (1-220), cm | 28.78, 20.00 |
| | Inundation duration (dur) | Continuous, (0.5-600), hours | 10.86, 3.00 |
| Contamination | Sewage (sew) | Binary, (Yes-1, No-0), - | -, - |
| | Garbage (gar) | | -, - |
| Warning | Warning (warn) | Binary, (Yes-1, No-0), - | -, - |
| Emergency measures | Pumped down water (pdw) | Binary, (Yes-1, No-0), - | -, - |
| | Temporary small-scale measures (temp) | | -, - |
| Private precautionary measures and flood experience | Acquire/purchase water barriers (w_bar) | Binary, (Yes-1, No-0), - | -, - |
| | Acquire/purchase pumping equipment (equ) | | -, - |
| | Elevated the house (elev) | | -, - |
| | Flood experience - floods per year in the last 10 years (fe) | Discrete, (0-4), - | -, 5.00 |
| Socio-economic/Building characteristics | Building area (barea) | Continuous, (12-1000), m$^2$ | 77.74, 60.00 |
| | Number of persons in the household (hh_size) | Continuous, (1-20), - | 4.90, 4.00 |
| | Years since the last renovation (renov) | Continuous, (0-10), years | 7.55, 3.00 |
| | Income (inc) | Discrete, (1-9), million VND (per month) | -, 3.00 |
| | Education (edu) | Binary, (Yes-1, No-0), - | -, - |
| | **Response variable** | | |
| Losses | Relative loss to building (rloss) | Continuous, (0-1), - | 0.16, 0.02 |

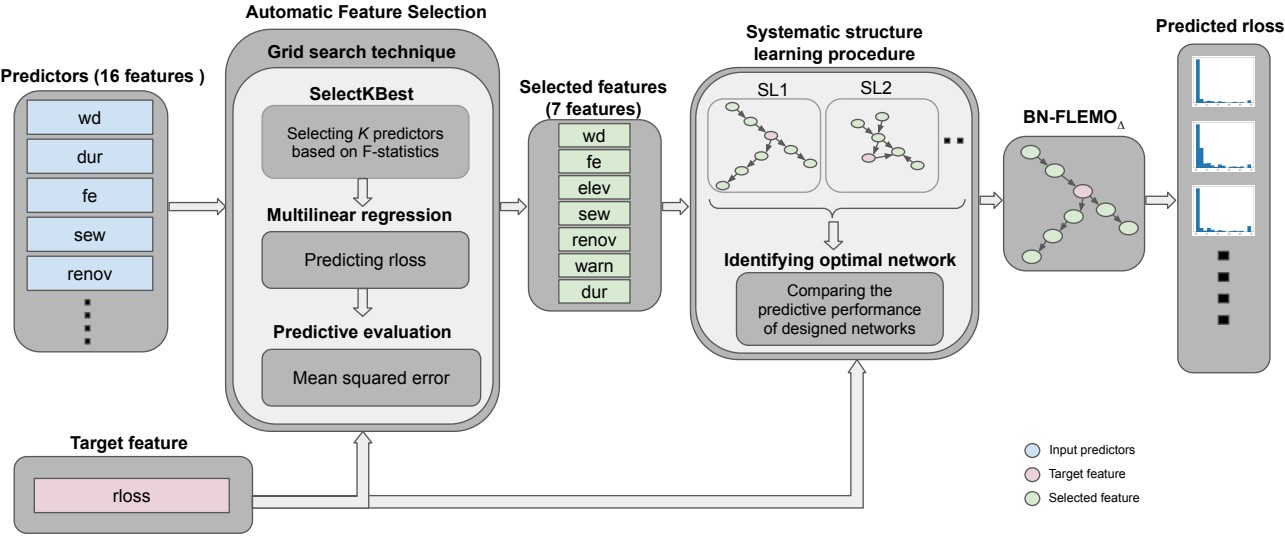

**Figure 1.** The proposed workflow to develop the flood loss estimation model. In the systematic structure learning procedure, SL1 and SL2 refer to the first and the second utilized structure learning algorithms.

tion workflow, we identify the optimal subset of features to use as input variables for the loss model and gain knowledge about

the loss processes in terms of the underlying relationship between the drivers and the relative loss.

### 2.2.2    Systematic Learning Process to Determine the Structure of the Bayesian Network

Bayesian Networks (BNs) are graphical models that describe the probabilistic dependencies between a set of random variables as a directed acyclic graph (Aguilera et al., 2011; Nagarajan et al., 2013). The graph consists of nodes representing random variables and arcs indicating conditional dependencies between the variables. Analogously, network variables that are not con-

nected are considered statistically independent. BNs can be used for both continuous and discrete variables, but in practice, BNs assume that all random variables are discrete (Chen et al., 2017). Thus, in this study, we design a discrete BN model to estimate flood loss. Thus, we discretize the continuous model variables using equal-frequency binning. We performed a sensitivity analysis regarding the discretization of relative loss, to determine how to effectively capture the variability of flood loss. This analysis specifically investigated the effects of discretizing relative loss into 3, 5, and 7 bins. Our findings indicated

that the alterations in the final BN structure were not substantial. Consequently, we opted for 5 bins, for all continuous variables, including relative loss, as they allow for a nuanced representation of patterns within the dataset (see Figure A2). Using factorization, the global joint probability distribution ($P(\mathbf{X}) = P(X_1, X_2, ..., X_k)$) of a random variable $\mathbf{X}$ can be written as:

$$P(\mathbf{X}) = \prod_{i=1}^{k} P(X_i \mid pa(X_i)), \tag{1}$$

where $k$ denotes the number of input features into the BN. $X_i$ and $pa(X_i)$ represent the $i$-th node and its corresponding parental node/nodes, respectively (i.e., the set of nodes pointing towards $X_i$) (Aguilera et al., 2011).

To build the BN-FLEMO$_\Delta$ structure, we utilize a data-driven learning approach in which 500 bootstrap samples are drawn from the data, and various structure learning algorithms are applied to generate potential Bayesian networks. These algorithms fall into three primary categories: score-based, constraint-based, and hybrid methods. In our study, we explore widely used structure learning approaches from each category Lüdtke et al. (2019); Schoppa et al. (2020). Specifically, we employ hill-climbing (hc)Russell (2010) as a score-based approach, incremental association (iamb)Tsamardinos et al. (2003) as a constraint-based approach, and max-min hill climbing (mmhc)Tsamardinos et al. (2006) along with general 2-phase restricted maximization (rs2max)Friedman et al. (2013) as hybrid approaches. Applying each method to the bootstrap samples results in 500 independent networks per algorithm. These realizations are then averaged, and arcs that appear frequently, based on a quantitative selection criterion, are retained in the final structure. The quantitative evaluation using mean absolute error (MAE), root mean squared error (RMSE), and mean continuous ranked probability score (Mean CRPS) indicates that the structure learned via the hill-climbing algorithm performs favorably compared to other approaches (see Figure A3). A qualitative assessment further supports this finding, showing that the hc-derived structure more accurately represents the underlying flood damage process (see Figure A5). Therefore, we selected the hc algorithm to construct the final BN-FLEMO$_\Delta$ structure. In addition to the network structure, the complete specification of the Bayesian network requires estimating the conditional probability tables (CPTs) for each node. For this purpose, we apply Bayesian parameter estimation using the Bayesian Dirichlet equivalent uniform (BDeu) score with an exact inference algorithm implemented in the bnlearn R package Scutari and Denis (2014). This method is used on the full empirical survey dataset, or relevant subsets during cross-validation, to ensure robust parameter learning.

## 2.3 Model Validation

### 2.3.1 Model Performance Comparison

To assess the predictive efficacy of BN-FLEMO$_\Delta$, we compare its performance with other flood loss modeling approaches. To make a fair model comparison, we compare our proposed probabilistic model with other probabilistic models, with the only exception of a deterministic stage damage function (SDF-Det), as this represents the state of the art (Merz et al., 2010; Scussolini et al., 2017; Couasnon et al., 2022). The uni- and multi-variable ML-based approaches include:

– **Stage damage functions (SDFs)**: The most commonly used flood loss modeling methods are deterministic SDFs, which are univariate techniques relying on water depth as a single flood loss driver. (Merz et al., 2010; Scussolini et al., 2017; Couasnon et al., 2022). Various studies have shown that the square root function provides more accurate estimates compared to linear relationships. Additionally, we use a probabilistic SDF (SDF-Prob) that utilizes probabilistic modeling techniques, such as Monte Carlo sampling, to estimate the probability distribution of flood damages (Schoppa et al., 2020). We formulate an SDF-Det that uses water depth as its independent variable as:

$$\mathbf{y} = \sqrt{\mathbf{wd}} \times c + \alpha, \tag{2}$$

where $\mathbf{wd} \in \mathbb{R}^{N \times 1}$ represents the measured water depth, and $c$ is an unknown coefficient. $\mathbf{y} \in \mathbb{R}^{N \times 1}$ expresses the relative loss values, and $\alpha$ denotes intercept. To transform SDF-Det into a probabilistic version, it necessitates defining a distribution function that can approximate the distribution of our target variable. Given that our target variable (relative loss) falls within the range of 0 to 1 and exhibits a bimodal distribution with peaks at 0 (no loss) and 1 (total loss), we opt for a zero-and-one-inflated Beta (**BEINF**) distribution to represent the target variable. The cumulative distribution function (CDF) of a zero-and-one-inflated Beta distribution can then be expressed as:

$$\mathbf{BEINF}(\mathbf{y}|\lambda, \gamma, \mu, \phi) = \lambda \mathbf{F}_{Bernoulli}(\mathbf{y}; \gamma) + (1 - \lambda)\mathbf{F}_{Beta}(\mathbf{y}; \mu, \phi)$$
$$logit(\mu) = \sqrt{\mathbf{wd}} \times c + \alpha, \tag{3}$$

where the CDF function $\mathbf{F}_{Bernoulli}$ represents a Bernoulli random variable with parameter $\gamma$, while $\mathbf{F}_{Beta}$ represents a Beta distribution with parameters $\mu$ (location) and $\phi$ (precision). In Equation (3), the values of $\lambda$, $\mu$, and $\gamma$ are constrained to the range $[0, 1]$, while $\phi$ must be greater than zero (Ospina and Ferrari, 2010). In this model formulation, $\mu$ varies for each household, whereas the other distribution parameters ($\lambda$, $\phi$, $\gamma$) remain constant across all households.

– **Random Forest (RF)**: RF is a powerful multi-variable ML tool in flood loss modeling due to its adeptness in handling intricate datasets and diverse predictor types effectively. This method is particularly advantageous for flood loss modeling tasks owing to its capability to accommodate both numerical and categorical variables commonly encountered in flood risk assessment data (Sieg et al., 2017). In flood loss modeling with RF, an ensemble of decision trees is constructed, with each tree trained on a random subset of the dataset. These decision trees individually predict flood losses based on various input variables. By amalgamating the predictions of multiple trees, RF can capture underlying relationships between predictors and the relative losses, thereby yielding accurate and dependable estimates. Notably, RF excels in handling nonlinear relationships and interactions between predictors, which is indispensable in flood risk assessment (Merz et al., 2013). Nonetheless, RF can be biased toward predictors with many possible splits. To address this issue, the conditional inference tree (CIT) algorithm based on permutation tests was developed, wherein RF can obtain conditional response distributions instead of mean values using quantile regression forest methodology (Hothorn et al., 2006). Recent studies increasingly have employed the conditional inference tree algorithm or a combination of conditional inference trees and quantile regression forests due to their advantages over conventional classification and regression tree algorithms (Sieg et al., 2017). The parameters controlling the RF model include the number of trees (ntree) and the number of randomly sampled predictors during partitioning (mtry). In our experiment we follow with common parameter values, typically set to ntree = 1,000 and mtry = 3 (Schoppa et al., 2020). While BN-FLMOE$_\Delta$ can update the parameters of the network (conditional probability distributions associated with each node) using new data, RF requires complete retraining with new data.

– **Bayesian regression (BR)**: BR is a statistical modeling technique that extends traditional linear regression by incorporating Bayesian principles. In BR, instead of estimating fixed model parameters, we treat them as random variables with probability distributions. This allows us to quantify uncertainty in our estimates and make probabilistic predic-

tions (Gelman et al., 1995). Thus, within flood loss modeling, we can adapt the BR concept to model relative loss with a zero-and-one-inflated beta distribution (Schoppa et al., 2020). This way we ensure that the model is capable of reproducing the extreme cases of no damage or total damage within our dataset. Although BR offers flexibility in modeling complex relationships and handling various types of uncertainties, making it a powerful tool in data analysis and prediction tasks, it does not provide a graphical representation to assist experts in analyzing the loss processes. Additionally, a BR models the conditional distribution of the relative loss given the predictor variables, while a BN computes the joint probability distribution over sets of variables using conditional probabilities (Mohor et al., 2021). In addition, unlike BN, which can handle missing input parameters, BR requires complete data and is prone to issues if missing inputs are not properly addressed.

For the probabilistic approaches SDF-Prob, BR, and BN-FLEMO$_\triangle$, we calculate the average of the distribution of predicted values to calculate the evaluation metrics.

The proposed feature selection workflow was implemented using Python 3.10 with the Scikit-learn 1.2.2 library. BN-FLEMO$_\triangle$ was implemented in R 4.2.2 using the Bnlearn 4.8.1 package. Notably, to ensure consistency across different components of the proposed framework, we used the same set of training and testing samples for each phase.

### 2.3.2 Evaluation Metrics:

To assess the performance of BN-FLEMO$_\triangle$ in comparison with other flood loss modeling approaches, we employ 10-fold cross-validation. This method involves iteratively training the model on nine subsets of the data and testing it on the remaining 10-*th* subset. This process is repeated 10 times to ensure that all data points are used for both training and testing. Then we report the average over these 10 runs in terms of the following metrics:

– Root mean squared error (RMSE): RMSE is a commonly used metric in regression analysis to measure the average magnitude of the errors between predicted and observed values. RMSE is calculated by taking the square root of the average of the squared differences between predicted and observed values. It provides a measure of the model's accuracy, with lower RMSE values indicating better performance.

– Mean absolute error (MAE): MAE is the average of the absolute difference between the predicted and observed values (residuals). MAE can be written as:

$$\frac{1}{N}||\mathbf{y} - \hat{\mathbf{y}}||_1 \tag{4}$$

– Mean bias error (MBE): MBE quantifies the average difference between predicted values and actual observations in a dataset. It measures the tendency of a model to consistently overestimate or underestimate the true values. MBE is calculated by taking the average of the differences between predicted and observed values, where positive values indicate an overall overestimation and negative values indicate an overall underestimation.

– Continuous ranked probability score (CRPS): CRPS is a metric used to evaluate probabilistic models by assessing the accuracy and reliability of their predictions. Unlike traditional point estimates, which provide only a single value, prob-

abilistic models generate entire distributions of possible outcomes. The CRPS compares these predicted distributions with observed outcomes, considering both the accuracy of the predicted values and the uncertainty represented by the distribution for each observation (Gneiting and Raftery, 2007), we can formulate CRPS as follows:

$$CRPS_i(\mathbf{F}_i, y) = \int_{-\infty}^{\infty} (\mathbf{F}_i(x) - \mathbf{1}_{\{x \geqslant y_i\}})^2 dx, \tag{5}$$

where $F_i(x)$ is the empirical CDF of the predictive distribution $f_i(x)$, and $\mathbf{1}\{.\}$ is the indicator function, which represents conditions or events in probability theory. We compute CRPS using an empirical CDF estimated from samples of $f_i(x)$. The CRPS ranges between 0 and 1, with the optimum at 0. Additionally, to facilitate comparison with other evaluation metrics, we computed the mean CRPS value in each cross-validation fold.

## 3 Results and Discussion

### 3.1 Important Variables for Flood Loss Modeling

Our proposed feature selection workflow automatically identifies the relevant features for estimating flood losses (see Figure 2). Figure 2 (a) illustrates that the optimal number of features to obtain the least MSE value ($\approx 0.0719$) is 7. The top 7 features with the highest scores are water depth (wd), flood experience (fe), building elevation (elev), sewage contamination (sew), years since last renovation (renov), warning (warn), and inundation duration (dur) (see Figure 2 (b)).

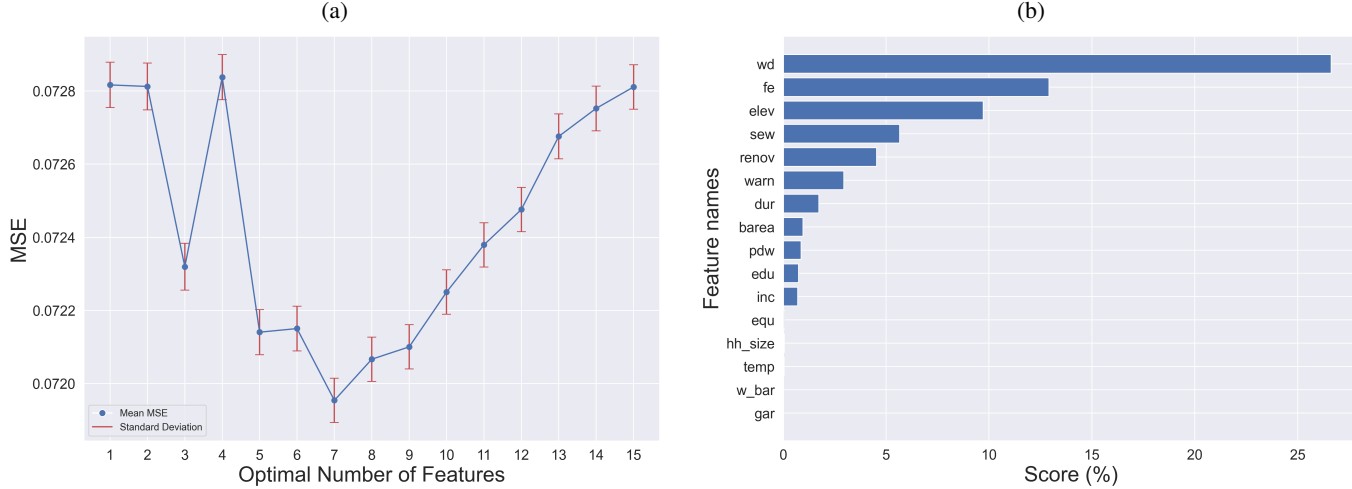

**Figure 2.** The proposed feature selection workflow (a) identifies the optimal number of features ($k$) and (b) provides the corresponding scores to each feature. The results are based on the mean of 10-fold cross-validation.

Water depth is found to be the most important driver for flood loss, which is in agreement with comprehensive reviews and state-of-the-art flood loss models, (Merz et al., 2013; Chinh et al., 2017; Rözer et al., 2019; Merz et al., 2010). The second

flood intensity parameter, i.e., inundation duration has been identified as an important predictor of flood loss as well, although it is only in the seventh position with a score of $\approx 1.72$.

The second important driver is flood experience. Flood experience has been found to influence households to implement precautionary measures such as elevating the building (Vishwanath Harish et al., 2022; Kreibich et al., 2005), which is the next important driver. It has been shown before, that it leads to flood loss reduction (Chinh et al., 2017). As another important factor, we can refer to sewage contamination. In several cases, contamination in flood water entering the buildings was found to increase the damage (Rözer et al., 2019; Thieken et al., 2005; Penning-Rowsell et al., 2014). Furthermore, the number of years since the building was last renovated, was also identified as an important factor. This finding also aligns with the fact that the building quality degenerates over time, recently renovated buildings are of better quality and were found to be more resistant to flooding (Chinh et al., 2017). Warning (i.e., whether, the household received a warning before the flood event) is also an important feature, which is a prerequisite for the implementation of emergency measures such as deploying sandbags and water barriers which reduce flood damage to buildings.

## 3.2 BN-FLEMO$_\Delta$

The trained Bayesian Network model using these 7 identified features elucidates the interactions across these features and their relationship to relative loss to residential buildings (see Figure 3). The direction of arrows in BN-FELMO$_\Delta$ indicates an association between two variables but does not necessarily imply causality (Lüdtke et al., 2019; Sairam et al., 2019). In the designed BN-FELMO$_\Delta$, the variable "rloss" has direct connections with water depth "wd", flood experience "fe" and years since last renovation "renov", which form the so-called Markov Blanket of rloss. In cases where the Markov Blanket is fully observed the other independent variables can be ignored. Thus, "wd", "fe" and "renov" are the most important predictors, which is aligned with flood loss dynamics and prior research findings (Chinh et al., 2017). Only, if observations of "wd", "fe" and "renov" are missing, observations on variables from outside the Markov Blanket provide knowledge helping to improve the prediction of "rloss". This is most likely the case in relation to "renov". Since it is hardly possible to know when the building was last renovated, but it is possible to observe whether the building is elevated (for instance by Google Street View (Pelizari et al., 2021)), the application of the loss model is improved in practice by the use of "elev". Elevating the building is one of the most common flood precautionary measures among households in HCMC (Vishwanath Harish et al., 2022). Interestingly, we also observe some differences to a relatively similar study by Chinh et al. (2017), in Can Tho City, Mekong Delta, who found a significantly higher importance of flood duration. These differences in flood processes and important input parameters for damage models confirm the need for region- and flood-type-specific loss models (Wagenaar et al., 2018; Mohor et al., 2021). While applying the BN-FLEMO$_\Delta$ (see Figure A6), known values are set to the predictor nodes, which updates the marginal probability distribution of the response variable conditioned on the predictors. When one or more values of nodes are unknown, the relative loss is conditioned on the values of the other known variables. Hence, the Bayesian network provides loss estimates even when some predictors are missing (Figure A6(b)). Figure A6 depicts the BN-FLEMO$_\Delta$ structure and parameters inferred from the survey data. In Figure A6 (a), the marginal probability distributions of the fitted network are presented. Furthermore, Figure A6 (b) showcases an exemplar prediction generated by BN-FLEMO$_\Delta$, utilizing three predictor variables presumed to be

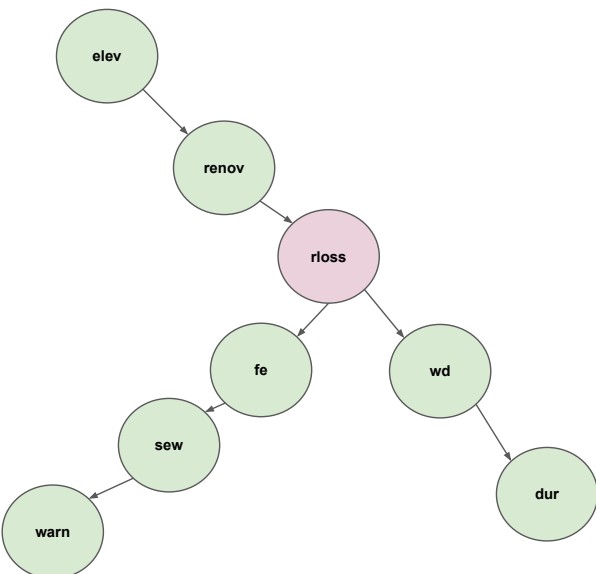

**Figure 3.** The final structure of BN-FLEMO$_\triangle$. The green nodes represent independent variables, including water depth (wd), flood experience (fe), years since the last renovation (renov), building elevation (elev), sewage contamination (sew), warning (warn) and inundation duration (dur). While the pink node presents the target variable, relative loss to residential buildings (rloss).

known (wd, fe, elev). The marginal probability distribution of the relative loss is updated conditionally based on the evidence in the nodes corresponding to these three predictors. The equal-frequency discretization of relative loss resulted in narrow bins for very low (less than 17%) losses so that these losses are precisely captured. This discretization reflects the damage processes in HCMC, which are characterized by frequent nuisance flooding resulting in rather low losses (Scheiber et al., 2023).

Furthermore, BN-FLEMO$_\triangle$ can provide information on the uncertainty involved in the flood loss estimation. Figure 4 illustrates predictive distributions for building loss of three randomly selected buildings. Vertical orange and green lines represent the actual observed relative loss and the average of the predictive distribution, respectively. However, it is evident that prediction accuracy and sharpness are greater for buildings with lower loss magnitudes (i.e., ID samples: 133 and 100) compared to the ones with more severe losses (i.e., ID sample: 800). Such an observation is due to the scarcity of extreme losses in the

dataset. As shown in Figure 4, BN-FLEMO$_\triangle$ can model the lower relative losses with less uncertainty, while in severe cases (i.e., complete damage: 1) the model has higher uncertainty.

In addition, to evaluate the performance of the proposed BN-FLEMO$_\triangle$ under the influence of different sets of input features, we designed an experiment where input features were incrementally added following the order determined by the feature selection process. As shown in Figure A4, using only the two initial loss-influencing variables (Set 2), namely wd and fe,

results in poor performance in terms of RMSE values. In contrast, Set 7, which includes seven identified features (wd, fe, elev, sew, renov, warn, dur), achieves relatively the best performance compared to other configurations. We also observe that

increasing the number of independent variables beyond this point does not improve predictive performance, highlighting the importance of the feature selection phase prior to the estimation process.

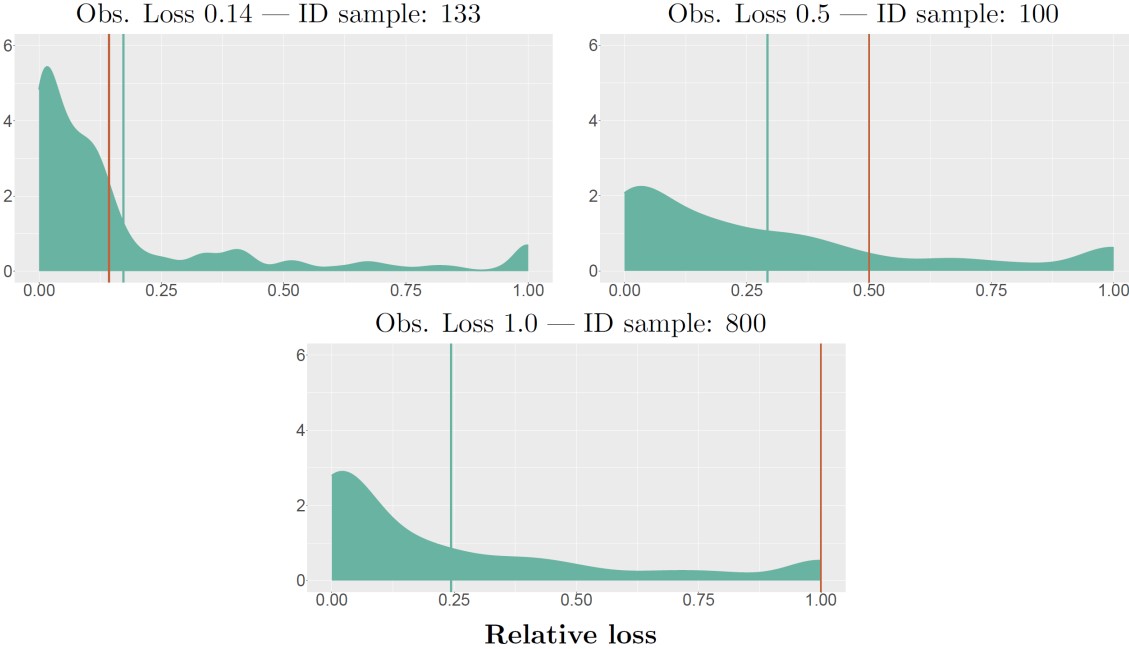

**Figure 4.** Samples of predictive densities of relative loss generated by BN-FLEMO$_\triangle$ for three randomly selected buildings (identified by ID sample). The observed loss is represented by orange, and green lines showcase the average of the predicted distribution generated by BN-FLEMO$_\triangle$. In this visualization, the x-axes represent the relative loss, while the y-axes display the density.

### 3.3 Model Validation

The quantitative evaluation of the model comparison, i.e., evaluation metrics, is illustrated in Figure 5. Our proposed BN-FLEMO$_\triangle$ performs better than the other ML-based approaches across nearly all error metrics, e.g., achieving the lowest absolute MAE value of 0.18±0.01 and the lowest Mean CRPS of 0.11±0.01. Results in terms of MBE, shown in Figure 5, reveal that BN-FLEMO$\triangle$ has a relatively lower bias compared to other ML-based approaches. BR and both stage-damage functions show a slightly lower bias, while RF tends to strongly overestimate the target value. In general, BR, SDF-Prob, SDF-
Det performed similarly well across all error metrics, whereas RF demonstrates the weakest performance of all approaches across all error metrics. Despite partly marginal differences between BN-FLEMO$_\triangle$ and BR, SDF-Prob, and SDF-Det, the advantages of our model are its applicability even with some missing input variables, the inherent quantification of predictive uncertainty and the graphical representation of the interaction between different flood loss-influencing factors and relative loss, offering insights into loss processes.

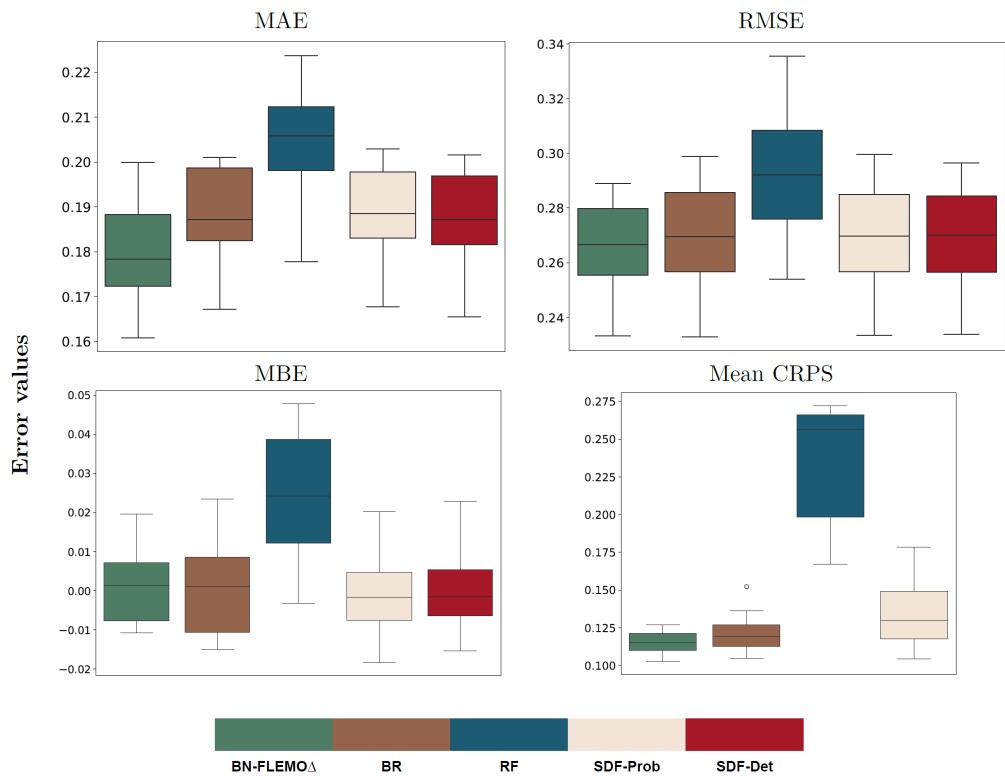

**Figure 5.** The quantitative evaluation of the predictive performance of ML-based approaches including, Bayesian network flood loss estimation model (BN-FLEMO$_\triangle$), Bayesian regression (BR), random forest (RF), and probabilistic stage damage function (SDF-Prob), and deterministic stage damage functions (SDF-Det). The x-axes display the deployed ML-based approaches, while the y-axes illustrate their respective error values based on various evaluation metrics.

## 4 Conclusion

The presented probabilistic flood loss estimation model BN-FLEMO$_\triangle$ is based on a large dataset (n=1000) of newly acquired empirical building-level survey data from HCMC. To construct this model, we introduce an automatic feature selection framework for the identification of key drivers of loss, complemented by a systematic learning approach for optimizing the Bayesian network structure to accurately capture loss processes. Notably, BN-FLEMO$_\triangle$ offers the capability to quantify model uncertainty by providing a probability distribution of losses, making it robust even in scenarios where data for certain predictors are missing. Moreover, the model incorporates predictors related to precautionary measures (e.g. building elevation), enabling the evaluation of adaptation strategies. Since Bayesian networks provide structured updating mechanisms with new data, BN-FLEMO$_\triangle$ is adaptable to changing conditions and transferable to other, similar delta cities. Consequently, it is a valuable tool for supporting decision-makers in developing adaptation strategies in data-scarce and rapidly evolving environments like delta

 cities. To this end, BN-FLEMO$_\triangle$ is provided for application by flood risk experts via the DECIDER Decision Support Tool (DST), complemented by descriptions and data . To ease the model application, a precomputed lookup table is provided, which associates all possible combinations of predictor variable values with the building loss that the Bayesian network predicts.

## Appendix A

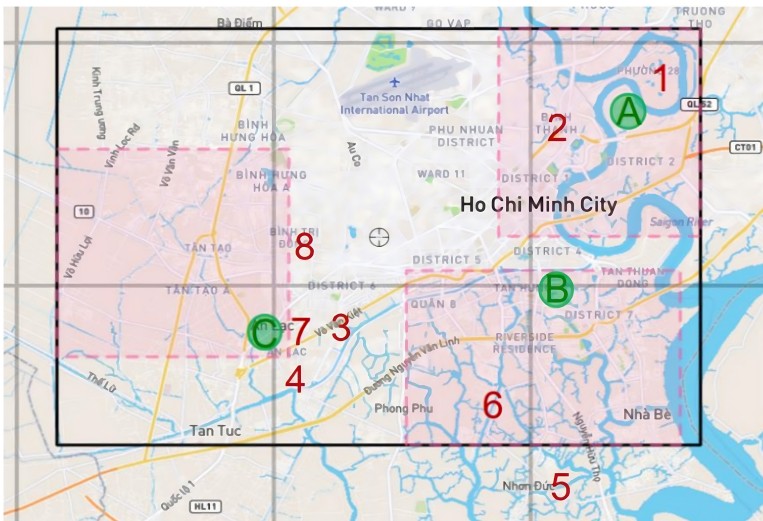

**Figure A1.** Selected survey areas in Ho Chi Minh City. Red numbers are the sites of the main survey in 2020. Green letters indicate the areas of the pre-test survey in December 2019 (Yang et al., 2020).

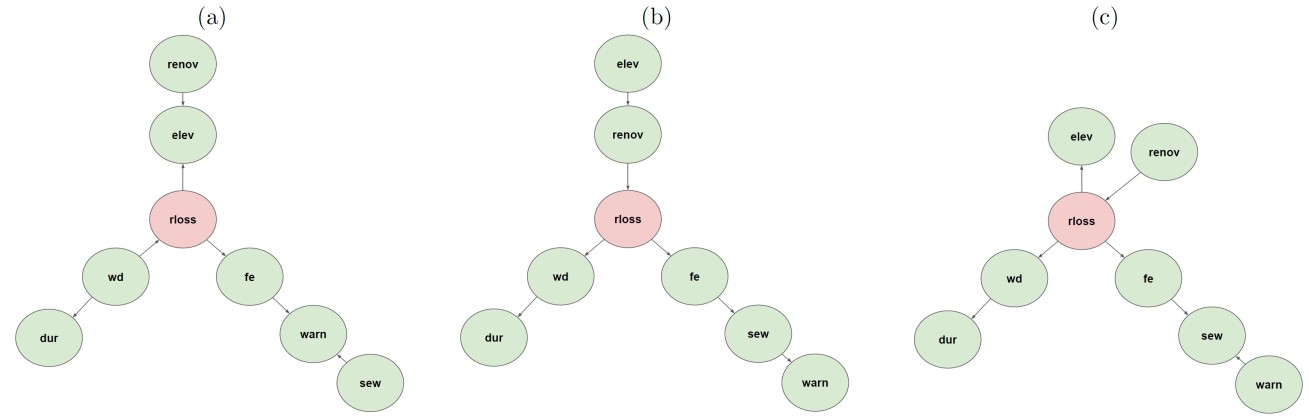

**Figure A2.** The constructed Bayesian networks by categorizing the relative loss (rloss) into, (a) 3 bins, (b) 5 bins, and (c) 7 bins.

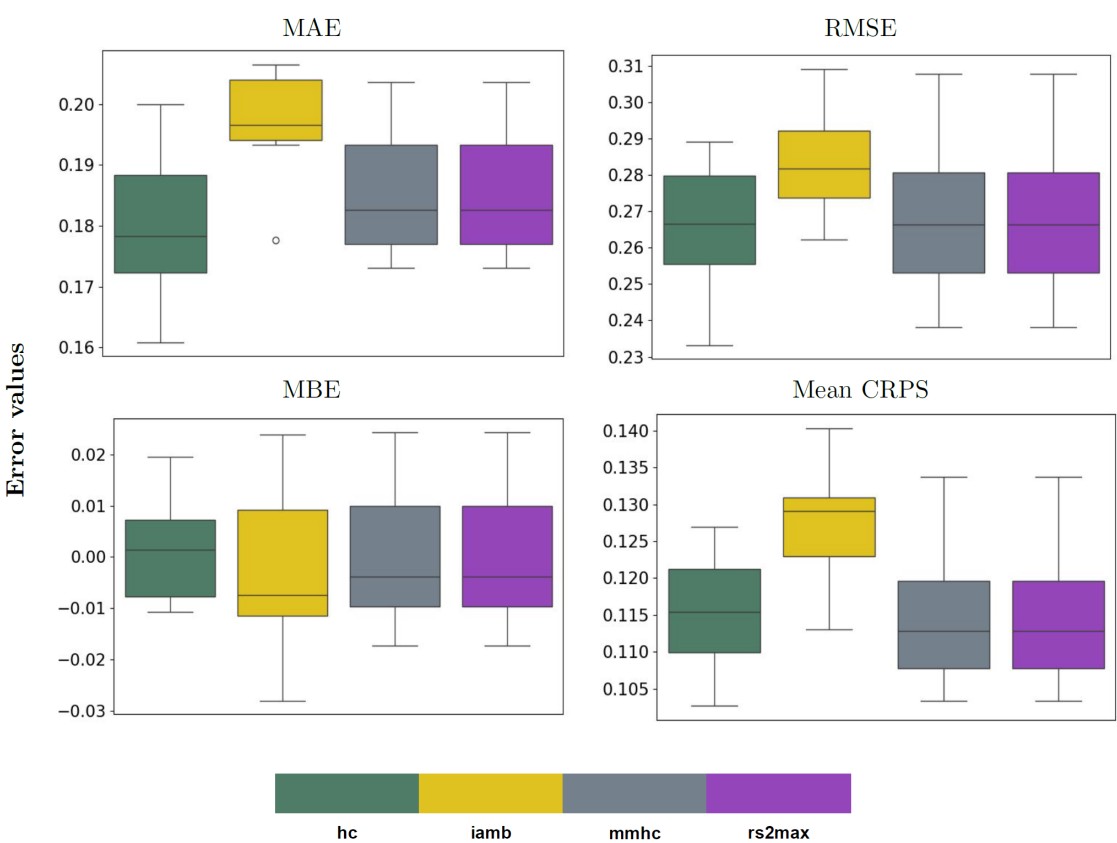

**Figure A3.** The quantitative performance of different structure learning algorithms, namely hill-climbing (hc); incremental association (iamb); max-min hill climbing (mmhc); and general 2-phase restricted maximization (rs2max), to build the BN-FLEMO$_\triangle$ model.

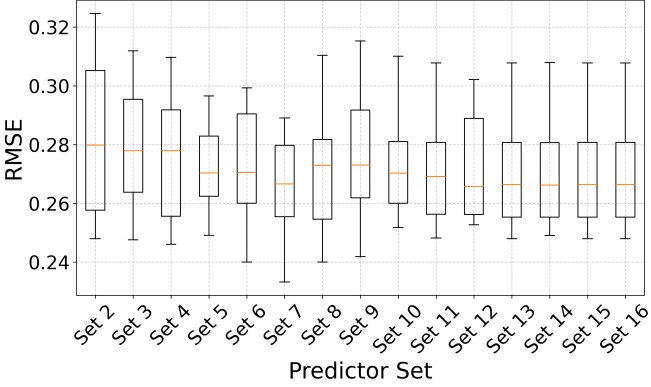

**Figure A4.** The figure illustrates the predictive performance of BN-FLEMO$_\triangle$ in terms of RMSE, with different predictor sets representing the incremental addition of independent variables as inputs to the model, following the order determined by the feature selection process.

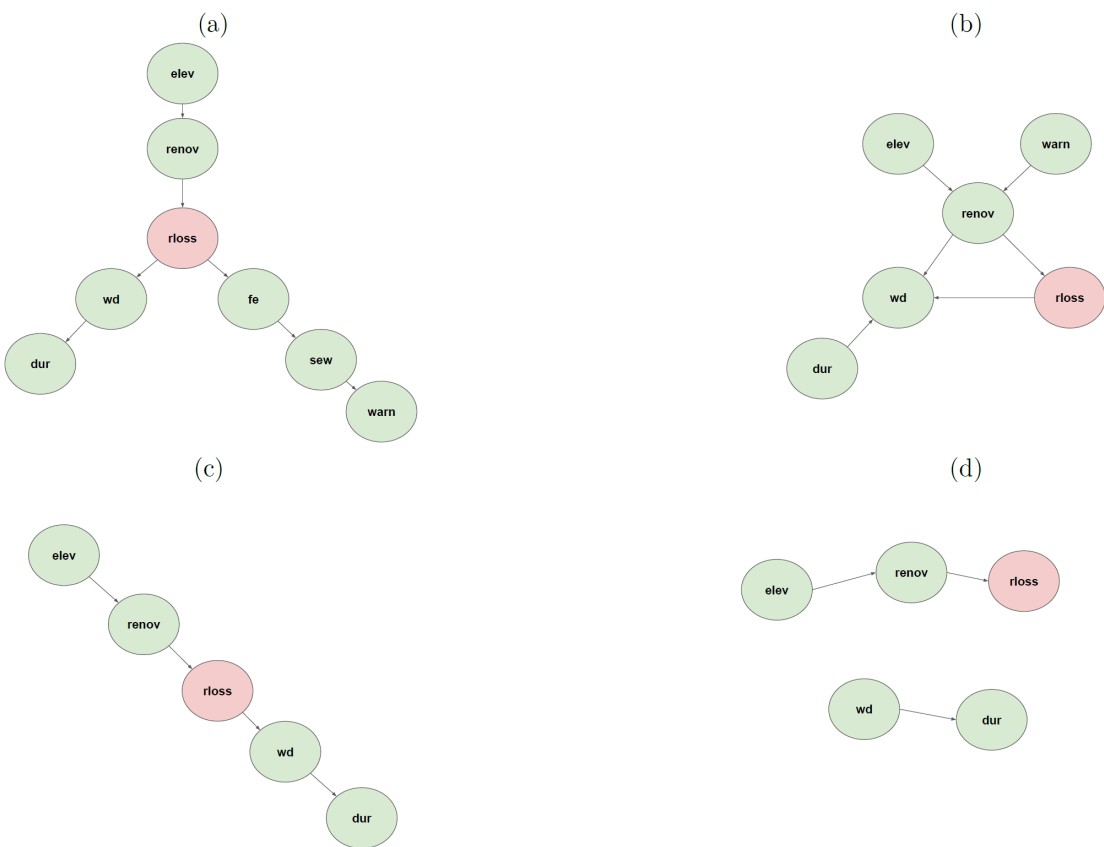

**Figure A5.** The Bayesian networks learned from (a) hc, (b) iamb, (c) mmhc, and (d) rs2max.

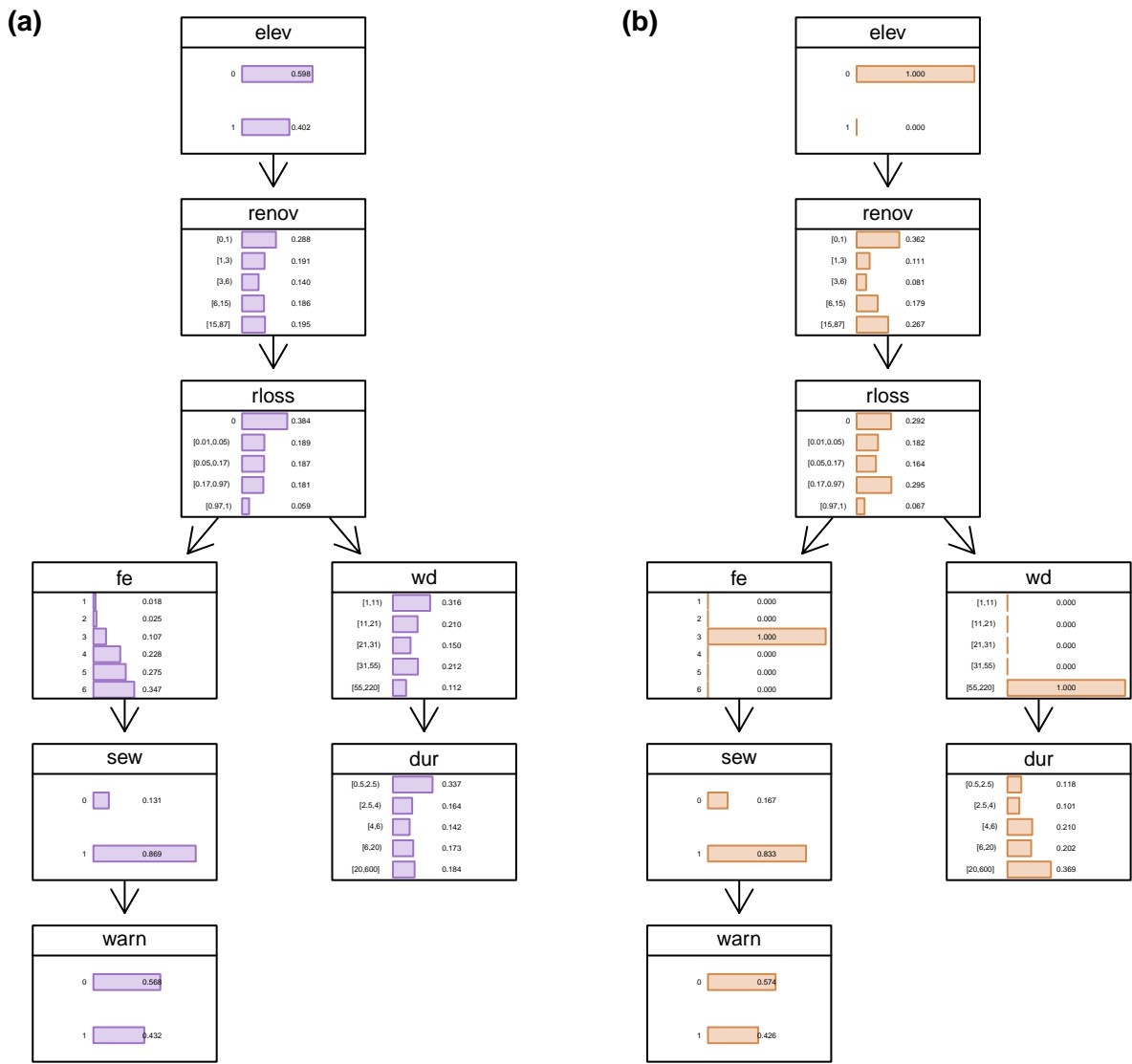

**Figure A6.** Bayesian network structure and parameters learned from the survey data. Panel (a) shows the marginal probability distributions of the fitted network. Panel (b) shows an example prediction of the Bayesian network, in which three predictor variables are assumed to be known (wd, fe, elev). The marginal probability distribution of the relative loss is updated conditional on the evidence in the nodes of these three predictors.

**Table A1.** The questionnaire that has been used during the survey campaign. This subset of the original version contains information on the 16 potentially important flood loss drivers studied in this work.

| Variable | Question | Optional (if applicable) |
|---|---|---|
| Water depth | What was the highest water point from your ground floor (cm)? | - |
| Inundation duration | What was the duration of inundation at the house (hours, minutes)? | - |
| Contamination | The flood water contained the following contaminants (multiple possible) | 0: No contamination (i.e., Normal rain/river water) <br> 1: Sewage water /excrements <br> 2: Daily living garbage |
| Warning | Did you receive a warning/know by yourself before the flood event? | 0: No <br> 1: Yes |
| Emergency Measures | Did you apply any emergency measures to prevent damages? (multiple answers possible) | 0: I did nothing <br> 1: Pumped down water <br> 2: Use sand bags, temporary and small-scale protection |
| Private Precautionary Measures | Which of the following precautionary measures have you implemented before the event? | 0: I did nothing <br> 1: Acquire/Purchase water barriers to prevent flooding in the house <br> 2: Acquire/Purchase pumping equipment to pump out water <br> 3: Elevate the house ground floor/foundation, etc. |
| Flood Experience | How many times have you been flooded since 10.2020 (i.e. flood water entering your house)? | 0: <= 1 (one or less than one) time per year <br> 1: > 1-2 times per year <br> 2: > 2-5 times per year <br> 3: > 5-10 times per year <br> 4: > 10 times per year |
| Building area | What is the floor size of the house in m$^2$? | - |
| Number of persons in the household | How many people are living in your household? | - |
| Years since the last renovation | If you performed a major renovation in the last 10 years, when was this renovation performed? (give the times of two major renovations if there were many) | - |
| Income | How high is the available income per month (million VND)? | 1: less than 1m <br> 2: 1m – 5m <br> 3: 5m – 10m <br> 4: 10m – 20m <br> 5: 20m – 30m <br> 6: 30m – 50m <br> 7: 50m – 80m <br> 8: 80m – 100m <br> 9: >100m |
| Education | Has any member of the household went to school? | 0. No <br> 1. Yes |
| Loss to building | How much did it cost you in total to repair your building (house/business)? | 1: ...million VND <br> 2: I did not repair anything |
| Reconstruction cost of the building | If you would rebuild your building (house/business) completely, what would this cost in million VND? Or, how much would it cost if your house is sold now (without land value)? | - |

*Code availability.* The code used for our analysis can be provided upon request

*Data availability.* The survey data used in this study will be made available via the HOWAS21 flood damage database. The HOWAS21 database is made accessible by Kreibich et al. (2007). In addition, precomputed lookup tables generated by BN-FLEMO$_\Delta$ can be found in Rafiezadeh Shahi et al. (2023). Such information facilitates the extension of flood loss estimation for the meso-scale application in HCMC.

*Author contributions.* Conceptualization: KRS, NS, LS, HK; Methodology: KRS, NS, LS, HK; Data curation: KRS, NS; Software: KRS, LS; Analysis and Visualization: KRS; Writing- original draft: KRS; Writing- review & editing: KRS, NS, LS, LeS, DT, HK.

*Competing interests.* The authors declare no competing interests.

*Acknowledgements.* This research has been supported by the German Federal Ministry of Education and Research (BMBF) within the framework of the DECIDER project (grant nos. 01LZ1703G and 01LZ1703A).

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
