# Peer review of "BN-FLEMO $_{\Delta}$ : A Bayesian Network-based Flood Loss Estimation Model for Adaptation Planning in Ho Chi Minh City, Vietnam"

_EGUsphere, 2024_

## Referee Comment (RC1)

The paper presents BN-FLEMOΔ, a probabilistic flood loss model tailored for residential buildings in delta cities like Ho Chi Minh City, using empirical survey data and machine learning. The model employs automatic feature selection and Discrete Bayesian Networks to capture probabilistic dependencies and provide a probabilistic distribution of losses.

The questions addressed by the paper are relevant and the paper presents a novel interesting dataset based on a survey study in Ho Chi Minh City.

Regarding the pipeline, the authors should clarify some major points, in particular, related to the feature selection step:

1) The authors use a 10-fold cross-validation for evaluating feature selection, the selection of the Bayesian Network (BN), and performance evaluation. However, it seems that they perform an initial 10-fold cross-validation (10FCV) for feature selection, and a separate one for model evaluation. This is evident from the fact that they select seven features based on the mean results of the first 10FCV, and also from the fact that they use Python and R—one for the first phase and one for the second (which would be acceptable if one could exactly map the splits from one language to the other). If this is the case, such an approach can lead to information leakage from the whole dataset (when selecting the features) into the model evaluation phase, since the feature selection step has effectively "seen" data from all folds. This can often yield overoptimistic estimates of the model's true performance on unseen data. The pipeline should perform feature selection within the same fold in which the model is trained, or find other ways to prevent information leakage.

2) Does the dataset have any missing values other than rloss? If so, the authors should specify how they handle these missing data points when computing the multivariate linear regression (MLR). MLR, by itself, cannot handle missing values. If the excluded features have missing values, their inclusion could negatively impact performance if they are set to a specific value (for example, 0). In general, it is important to clarify how these particular data points are handled.

3) The strategy chosen for feature selection is unusual when combined with Bayesian networks. The authors evaluate the importance of each individual feature using F-statistics to measure the relationship between the variable and the target variable. This univariate approach can overlook the signal from variables that, when combined with another feature, a Bayesian network (BN) might capture. Moreover, the hyperparameter k is chosen using a linear model. These choices are not strictly speaking incorrect, but they may reduce the predictive capacity of the entire pipeline. Additionally, in BNs, it is possible to reduce the complexity of the model during the structural learning phase in ways that preserve the multivariate nature of the problem.

    a) The dataset has only 16 variables; is feature selection necessary? BN structural learning can usually handle this number of variables with an appropriate choice of algorithms and parameters, without the need for prior selection.

    b) Feature selection is part of the pipeline proposed in the paper. The authors should compare its performance not only with a model trained on the subset of selected variables but also with models trained on the entire dataset. For example, they should compare it with a random forest trained on all the features, since RF can easily handle 16 variables.

    c) To demonstrate the significance of feature selection, the authors should compare the pipeline's performance with BN models trained on all the features, using appropriate algorithms that can reduce complexity or identify Markov blankets from the data (see, e.g., Vogel 2018 and the BN literature).

    d) To better illustrate the impact of feature selection, it would be insightful to include a plot similar to Figure 2a, but showing the performance of BNs

trained with varying numbers of features. This could also be done for only the best model identified by the authors. If computational limitations exist, it would be reasonable to limit the study to a few additional features beyond seven.

e) It would be insightful to see the error bars with the standard deviation over the 10-fold in Figure 2a.

Regarding the training:

4) In line 226 they should specify what exact Bayesian algorithm they are using.

Regarding the dataset:

5) To better present the dataset, it would be useful to include the number of missing values for each feature, if any, in Table 1 or another section of the paper.

6) It would be useful to see the histogram of rloss before the discretization.

Besides these issues that need to be addressed, the paper is well-written, and the overall presentation is sufficiently well-structured. The lookup table provided in the supplemental material is useful and adequately described.

---

## Referee Comment (RC2)

Journal: NHESS
Title: **BN-FLEMOΔ: A Bayesian Network-based Flood Loss Estimation Model for Adaptation Planning in Ho Chi Minh City, Vietnam**
Author(s): Shahi al.
MS No.: egusphere-2024-3631
MS Type: Research Article
**Iteration: First review**

The manuscript presents a new probabilistic multi-variable loss model for residential buildings in Ho Chi Minh City (HCMC). The model is based on a set of about 1000 newly collected loss data and Bayesian Network methods. The topic fits within the scope of the journal. The structure of the paper is clear, the methodology and results are clearly explained, and results are supported by data. In my opinion, the paper can be accepted for publication after minor points are clarified/addressed.

**Minor concerns:**

Pg. 4 "Nevertheless, 467 out of 1530 data points contained missing values in flood loss predictors" → Do authors refer to water depth?

Table 1 → Could authors better explain how these 16 variables have been selected? And why physical vulnerability of buildings (e.g. building structure) has not been considered? Could all buildings in the affected considered similar from this perspective?

Pg. 11 "In comparison with studies conducted in Europe (Kreibich et al., 2017; Wagenaar et al., 2018; Mohor et al., 2021), we observe significantly higher importance of renovation and elevation of the building, as to our knowledge these variables have not been identified as relevant loss-influencing variables there" → I do not agree with this sentence. Many loss models include building elevation and level of maintenance/renovation as independent variables.

Pg. 13 "Consequently, it is a valuable tool for supporting decision-makers in developing adaptation strategies in data-scarce and rapidly evolving environments like delta cities" → Could authors supply some examples of how the models can be used in practice by decision makers? I am afraid that this kind of models are hardly transferable to decision makers.

---

## Author Comment (AC1)

**Dear Editor,**

**Please find enclosed, the revised manuscript entitled "BN-FLEMO$_\triangle$: A Bayesian Network-based Flood Loss Estimation Model for Adaptation Planning in Ho Chi Minh City, Vietnam". All the comments/suggestions raised by the reviewers have been carefully addressed in the revised version. Below, all our replies are given in bold typeface. Moreover, all the changes have been highlighted in blue in the revised manuscript.**

**We would like to take this opportunity to express our appreciation for the work of the Editor, the anonymous Associate Editor, and two Reviewers who provided very relevant and constructive comments and suggestions. We have addressed all the requests and amended the manuscript accordingly. The provided comments and suggestions led us to considerably improve the manuscript.**

- **Reviewer 2:**

  The manuscript presents a new probabilistic multi-variable loss model for residential buildings in Ho Chi Minh City (HCMC). The model is based on a set of about 1000 newly collected loss data and Bayesian Network methods. The topic fits within the scope of the journal. The structure of the paper is clear, the methodology and results are clearly explained, and results are supported by data. In my opinion,the paper can be accepted for publication after minor points are clarified/addressed.

  **Response: We appreciate Reviewer 2 for his/her valuable comments which helped us to enhance the quality of the manuscript.**

  **Minor concerns**:
  1. Page. 4 "Nevertheless, 467 out of 1530 data points contained missing values in flood loss predictors" → Do authors refer to water depth?

     **Response: We clarify that this statement refers to missing values in any of the predictors used for flood loss estimation, not just water depth. The text has been revised to remove any ambiguity.**

     Households were asked to report on two flood events (i.e., the most recent event and the most serious event in terms of impact within the last 10 years). Out of 1000 households, 530 provided information on both types of events, while the remaining households reported only one event (in these cases, the recent event was also the most serious). This resulted in 1530 records of flood loss data. Among these records, 467 contained missing values in one or more of the flood loss predictors (e.g., water

depth, inundation duration) or target variable (rloss). To ensure the integrity of the analysis, we adopted a complete-case approach by excluding all records with missing values. Subsequently, 16 loss-influencing variables were selected based on an extensive literature review and consultations with domain experts in Ho Chi Minh City (Table 1).

2. Table 1 → Could authors better explain how these 16 variables have been selected? And why physical vulnerability of buildings (e.g. building structure) has not been considered? Could all buildings in the affected considered similar from this perspective?

**Response: The 16 variables were selected based on an extensive literature review and consultations with domain experts in Ho Chi Minh City. Although physical vulnerability factors, such as building structure, are important, our dataset does not include this information for the study area.**

3. Page. 11 "In comparison with studies conducted in Europe (Kreibich et al., 2017; Wagenaar et al., 2018; Mohor et al., 2021), we observe significantly higher importance of renovation and elevation of the building, as to our knowledge these variables have not been identified as relevant loss-influencing variables there" → I do not agree with this sentence. Many loss models include building elevation and level of maintenance/renovation as independent variables.

**Response: We acknowledge that several European studies include building elevation and renovation as predictors. However, our analysis indicates that these factors have an even more pronounced influence in the context of HCMC. That said, to avoid any potential confusion, we have removed the sentence from the manuscript.**

4. Page. 13 "Consequently, it is a valuable tool for supporting decision-makers in developing adaptation strategies in data-scarce and rapidly evolving environments like delta cities" → Could authors supply some examples of how the models can be used in practice by decision makers? I am afraid that this kind of models are hardly transferable to decision makers.

**Response: We appreciate the reviewer's feedback. We agree that highlighting practical applications is essential. This is precisely why approaches like the proposed BN-FLEMO$_\triangle$ are valuable, they not only provide a graphical representation that facilitates communication with decision-makers and stakeholders but also allow users to verify whether the results align with physical flood processes (e.g., understanding the relationships between independent and dependent variables, as illustrated in Figure 1). Furthermore, as noted in the conclusion, the model quantifies uncertainty, enabling users to incorporate this information**

[Figure]

Fig. 1: The Bayesian networks learned from (a) hc, (b) iamb, (c) mmhc, and (d) rs2max.

**into the decision-making process. To further demonstrate the practical application of BN-FLEMO$_\triangle$, the model has also been made available to flood risk experts through the DECIDER Decision Support Tool. We have now included this additional information in the manuscript as follows:**

To this end, BN-FLEMO$_\triangle$ is provided for application by flood risk experts via the DECIDER Decision Support Tool (DST), complemented by descriptions and data . To ease the model application, a precomputed lookup table is provided, which associates all possible combinations of predictor variable values with the building loss that the Bayesian network predicts.

---

## Author Comment (AC2)

**Dear Editor,**

**Please find enclosed, the revised manuscript entitled "BN-FLEMO$_\triangle$: A Bayesian Network-based Flood Loss Estimation Model for Adaptation Planning in Ho Chi Minh City, Vietnam". All the comments/suggestions raised by the reviewers have been carefully addressed in the revised version. Below, all our replies are given in bold typeface. Moreover, all the changes have been highlighted in blue in the revised manuscript.**

**We would like to take this opportunity to express our appreciation for the work of the Editor, the anonymous Associate Editor, and two Reviewers who provided very relevant and constructive comments and suggestions. We have addressed all the requests and amended the manuscript accordingly. The provided comments and suggestions led us to considerably improve the manuscript.**

- **Reviewer 1:**

  The paper presents BN-FLEMO$_\triangle$, a probabilistic flood loss model tailored for residential buildings in delta cities like Ho Chi Minh City, using empirical survey data and machine learning. The model employs automatic feature selection and Discrete Bayesian Networks to capture probabilistic dependencies and provide a probabilistic distribution of losses. The questions addressed by the paper are relevant and the paper presents a novel interesting dataset based on a survey study in Ho Chi Minh City. Regarding the pipeline, the authors should clarify some major points, in particular, related to the feature selection step.

  **Response: We appreciate Reviewer 1 for her/his valuable comments which helped us to enhance the quality of the manuscript.**

  1. The authors use a 10-fold cross-validation for evaluating feature selection, the selection of the Bayesian Network (BN), and performance evaluation. However, it seems that they perform an initial 10-fold cross-validation (10FCV) for feature selection, and a separate one for model evaluation. This is evident from the fact that they select seven features based on the mean results of the first 10FCV, and also from the fact that they use Python and R—one for the first phase and one for the second (which would be acceptable if one could exactly map the splits from one language to the other). If this is the case, such an approach can lead to information leakage from the whole dataset (when selecting the features) into the model evaluation phase, since the feature selection step has effectively "seen" data from all folds. This can often yield overoptimistic estimates of the model's true performance on unseen data. The pipeline should perform feature selection within the same fold in which the model is trained, or find other ways to prevent information leakage.

**Response: We thank the reviewer for her/his detailed and constructive comment. As the reviewer rightly pointed out, ensuring consistency in the utilized data across different programming languages and frameworks (i.e., feature selection and monetary loss estimation) is crucial to avoid overfitting. To address this, we have already implemented a workflow in the current version, where the indices of the training/testing samples are stored during the initial phase (i.e., feature selection) and subsequently used in later phases. Accordingly, we have amended the manuscript to clarify this, as follows:**

Line 219-220: Notably, to ensure consistency across different components of the proposed workflow, we used the same set of training and testing samples for each phase.

2. Does the dataset have any missing values other than rloss? If so, the authors should specify how they handle these missing data points when computing the multivariate linear regression (MLR). MLR, by itself, cannot handle missing values. If the excluded features have missing values, their inclusion could negatively impact performance if they are set to a specific value (for example, 0). In general, it is important to clarify how these particular data points are handled.

**Response: Indeed, MLR is not designed to handle missing values, which is why we have already excluded 467 out of 1530 data records that contained missing values in the various predictors of flood loss. For clarification, we have explicitly stated and highlighted this in the "Household Survey Data" subsection, lines 89 and following, as follows:**

Households were asked to report on two flood events (i.e., the most recent event and the most serious event in terms of impact within the last 10 years). Out of 1000 households, 530 provided information on both types of events, while the remaining households reported only one event (in these cases, the recent event was also the most serious). This resulted in 1530 records of flood loss data. Among these records, 467 contained missing values in one or more of the flood loss predictors (e.g., water depth, inundation duration) or target variable (rloss). To ensure the integrity of the analysis, we adopted a complete-case approach by excluding all records with missing values. Subsequently, 16 loss-influencing variables were selected based on an extensive literature review and consultations with domain experts in Ho Chi Minh City (Table 1).

3. The strategy chosen for feature selection is unusual when combined with Bayesian networks. The authors evaluate the importance of each individual feature using F-statistics to measure the relationship between the variable and the target variable. This

univariate approach can overlook the signal from variables that, when combined with another feature, a Bayesian network (BN) might capture. Moreover, the hyperparameter k is chosen using a linear model. These choices are not strictly speaking incorrect, but they may reduce the predictive capacity of the entire pipeline. Additionally, in BNs, it is possible to reduce the complexity of the model during the structural learning phase in ways that preserve the multivariate nature of the problem.

**Response: The primary goal of the deployed feature selection is to automate the process of identifying relevant features that influence monetary loss. As the reviewer indicated, the first phase of feature selection, where the F-statistic evaluates the dependency of each predictor on the target variable, may overlook multivariate relationships between subsets of predictors and the target variable. To address this limitation, we have implemented a second phase in the feature selection pipeline, where we assess the overall performance of the top $k$ selected features, ensuring that multivariate relationships are considered. This approach allows us not only to examine the relationship between individual predictors and the target variable but also to evaluate the significance of predictor subsets in estimating loss.**

a. The dataset has only 16 variables; is feature selection necessary? BN structural learning can usually handle this number of variables with an appropriate choice of algorithms and parameters, without the need for prior selection.

   **Response: Although the number of input features is 16, some features contain redundant information that does not contribute to assessing flood loss. As shown in Figure 1, automatic feature selection significantly enhances both model interpretability and predictive performance. By focusing on the key drivers of flood loss, the model's complexity is reduced, making the resulting BN structure more transparent. This, in turn, facilitates a better understanding of flood loss processes.**

b. Feature selection is part of the pipeline proposed in the paper. The authors should compare its performance not only with a model trained on the subset of selected variables but also with models trained on the entire dataset. For example, they should compare it with a random forest trained on all the features, since RF can easily handle 16 variables.

   **Response: We appreciate this suggestion. In fact, during our experiments, we compared the proposed feature selection approach with standard methods such as RF. However, as shown in Figure 2, our analysis revealed that the features selected by RF do not fully align with the physical flood process.**

**For instance, water depth was not identified as the primary driver of relative loss, and features such as the number of people in the household appeared as prominent, while precautionary measures like elevating the house were deemed less important. Additionally, RF does not automatically determine the optimal number of features. For these reasons, we decided to exclude this result from the main manuscript. However, to address the reviewer's comment, we have included this result in our response.**

c. To demonstrate the significance of feature selection, the authors should compare the pipeline's performance with BN models trained on all the features, using appropriate algorithms that can reduce complexity or identify Markov blankets from the data (see, e.g., Vogel 2018 and the BN literature).

**Response: Indeed, similar to the approach in [1], where the authors employed two types of graphical models, BNs and Markov blankets, to identify the most relevant variables for flood damage prediction, we also applied constraint-based methods to construct our proposed BNs using a selected set of independent variables (as highlighted in blue below). However, the key distinction between our approach and that of [1] lies in the method of selecting independent variables. In our study, we utilize an automated feature selection pipeline that systematically evaluates the impact of each independent variable on the target variable. In contrast, [1] identified important predictors based on the frequency of occurrence of each variable across multiple algorithms and a dataset comprising six distinct flood events. Furthermore, in response to your subsequent comment regarding the impact of feature selection, we have compared and discussed the performance of the BN-FLEMO$_\triangle$ model using different sets of features in the corresponding response.**

To build the BN-FLEMO$_\triangle$ structure, we utilize a data-driven learning approach in which 500 bootstrap samples are drawn from the data, and various structure learning algorithms are applied to generate potential Bayesian networks. These algorithms fall into three primary categories: score-based, constraint-based, and hybrid methods. In our study, we explore widely used structure learning approaches from each category [2], [3]. Specifically, we employ hill-climbing (hc)[4] as a score-based approach, incremental association (iamb)[5] as a constraint-based approach, and max-min hill climbing (mmhc)[6] along with general 2-phase restricted maximization (rs2max)[7] as hybrid approaches. Applying each method to the bootstrap samples results in 500 independent networks per algorithm. These realizations are then averaged, and arcs that appear frequently, based on a quantitative selection criterion, are retained in the final structure. The quantitative evaluation using mean absolute error (MAE), root mean squared error (RMSE), and mean continuous

ranked probability score (Mean CRPS) indicates that the structure learned via the hill-climbing algorithm performs favorably compared to other approaches (see Figure 3).

d. To better illustrate the impact of feature selection, it would be insightful to include a plot similar to Figure 2a, but showing the performance of BNs trained with varying numbers of features. This could also be done for only the best model identified by the authors. If computational limitations exist, it would be reasonable to limit the study to a few additional features beyond seven.

**Response: We agree that this visualization is insightful. Figure 4 presents the performance of BN models with varying numbers of predictors, following the order determined by the feature selection process. The plot clearly demonstrates that optimal performance is achieved with a subset of seven features, while including too many features leads to a decline in performance. Accordingly, we have added the figure, along with the following information, to the revised version of the manuscript.**

In addition, to evaluate the performance of the proposed BN-FLEMO$_\Delta$ under the influence of different sets of input features, we designed an experiment where input features were incrementally added following the order determined by the feature selection process. As shown in Figure 4, using only the two initial loss-influencing variables (Set 2), namely wd and fe, results in poor performance in terms of RMSE values. In contrast, Set 7, which includes seven identified features (wd, fe, elev, sew, renov, warn, dur), achieves relatively the best performance compared to other configurations. We also observe that increasing the number of independent variables beyond this point does not improve predictive performance, highlighting the importance of the feature selection phase prior to the estimation process.

e. It would be insightful to see the error bars with the standard deviation over the 10-fold in Figure 2a.

**Response: As suggested, we have updated Figure 2a to include error bars representing the standard deviation from the 10-fold cross-validation. This addition offers deeper insight into the variability and robustness of the feature selection process.**

4. Regarding the training, in line 226, they should specify what exact Bayesian algorithm they are using.

**Response: As commented by the reviewer, we have now clarified the specific Bayesian inference method used for estimating conditional probabilities in the network. Specifically, we applied Bayesian parameter estimation using the BDeu score with an exact inference algorithm, as implemented in the bnlearn R package [8]. The revised version is as follows:**

A qualitative assessment further supports this finding, showing that the hc-derived structure more accurately represents the underlying flood damage process (see Figure 5). Therefore, we selected the hc algorithm to construct the final BN-FLEMO$_\Delta$ structure. In addition to the network structure, the complete specification of the Bayesian network requires estimating the conditional probability tables (CPTs) for each node. For this purpose, we apply Bayesian parameter estimation using the Bayesian Dirichlet equivalent uniform (BDeu) score with an exact inference algorithm implemented in the bnlearn R package [8]. This method is used on the full empirical survey dataset, or relevant subsets during cross-validation, to ensure robust parameter learning.

5. Regarding the dataset, to better present the dataset, it would be useful to include the number of missing values for each feature, if any, in Table 1 or another section of the paper.

   **Response: As mentioned in our response to the second comment raised by the reviewer, we have already excluded 467 out of 1530 data points that contained missing values in the various predictors of flood loss. Therefore, the existing table only represents data records where both the predictors and the target variable contain no missing information. For clarity, we have already indicated and highlighted this in the "Household Survey Data" subsection.**

6. Regarding the dataset, it would be useful to see the histogram of rloss before the discretization.

   **Response: We have now included a histogram of the rloss values (before discretization) in the revised manuscript (see new Figure 6 in the supplementary material). This visualization provides insight into the distribution of the target variable and supports our choice of discretization strategy.**

7. Besides these issues that need to be addressed, the paper is well-written, and the overall presentation is sufficiently well-structured. The lookup table provided in the supplemental material is useful and adequately described.

   **Response: We thank the reviewer for the positive feedback and are pleased that the overall presentation was appreciated.**

[Figure]

Fig. 1: The proposed feature selection workflow (a) identifies the optimal number of features ($k$) and (b) provides the corresponding scores to each feature. The results are based on the mean of 10-fold cross-validation.

[Figure]

Fig. 2: The figure illustrates feature importance as determined by the random forest (RF) model, which uses the mean decrease in impurity for this process.

REFERENCES

[1] K. Vogel, L. Weise, K. Schröter, and A. H. Thieken, "Identifying driving factors in flood-damaging processes using graphical models," *Water resources research*, vol. 54, no. 11, pp. 8864–8889, 2018.
[2] S. Lüdtke, K. Schröter, M. Steinhausen, L. Weise, R. Figueiredo, and H. Kreibich, "A Consistent Approach for Probabilistic Residential Flood Loss Modeling in Europe," *Water Resources Research*, vol. 55, pp. 10616–10635, dec 2019.
[3] L. Schoppa, T. Sieg, K. Vogel, G. Zöller, and H. Kreibich, "Probabilistic flood loss models for companies," *Water Resources Research*, vol. 56, no. 9, p. e2020WR027649, 2020.
[4] S. J. Russell, *Artificial intelligence a modern approach*. Pearson Education, Inc., 2010.
[5] I. Tsamardinos, C. Aliferis, and A. Statnikov, "Algorithms for large scale markov blanket discovery," pp. 376–381, 01 2003.
[6] I. Tsamardinos, L. E. Brown, and C. F. Aliferis, "The max-min hill-climbing bayesian network structure learning algorithm," *Machine learning*, vol. 65, pp. 31–78, 2006.
[7] N. Friedman, I. Nachman, and D. Pe'er, "Learning bayesian network structure from massive datasets: The" sparse candidate" algorithm," *arXiv preprint arXiv:1301.6696*, 2013.
[8] M. Scutari and J.-B. Denis, *Bayesian Networks*. Boca Raton: Chapman and Hall/CRC, jun 2014.

[Figure]

Fig. 3: The quantitative performance of different structure learning algorithms, namely hill-climbing (hc); incremental association (iamb); max-min hill climbing (mmhc); and general 2-phase restricted maximization (rs2max), to build the BN-FLEMO$_\Delta$ model.

[Figure]

Fig. 4: The figure illustrates the predictive performance of BN-FLEMO$_\Delta$ in terms of RMSE, with different predictor sets representing the incremental addition of independent variables as inputs to the model, following the order determined by the feature selection process.

[Figure]

Fig. 5: The Bayesian networks learned from (a) hc, (b) iamb, (c) mmhc, and (d) rs2max.

[Figure]

Fig. 6: The histogram depicts the distribution of relative loss (rloss) values across different data points.